# Powder Bed Fusion Additive Manufacturing Using Critical Raw Materials: A Review

**DOI:** 10.3390/ma14040909

**Published:** 2021-02-14

**Authors:** Vladimir V. Popov, Maria Luisa Grilli, Andrey Koptyug, Lucyna Jaworska, Alexander Katz-Demyanetz, Damjan Klobčar, Sebastian Balos, Bogdan O. Postolnyi, Saurav Goel

**Affiliations:** 1Israel Institute of Metals, Technion R&D Foundation, Haifa 3200003, Israel; kalexand@trdf.technion.ac.il; 2ENEA–Italian National Agency for New Technologies, Energy and Sustainable Economic Development, Energy Technologies and Renewable Sources Department, Casaccia Research Centre, Via Anguillarese 301, 00123 Rome, Italy; marialuisa.grilli@enea.it; 3SportsTech Research Center, Mid Sweden University, Akademigatan 1, SE-83125 Östersund, Sweden; andrey.koptyug@miun.se; 4Faculty of Non-Ferrous Metals, AGH University of Science and Technology, 30-059 Krakow, Poland; ljaw@agh.edu.pl; 5Faculty of Mechanical Engineering, University of Ljubljana, Aškerčeva c. 6, 1000 Ljubljana, Slovenia; damjan.klobcar@fs.uni-lj.si; 6Department of Production Engineering, Faculty of Technical Science, University of Novi Sad, Novi Sad, Trg Dositeja Obradovica 6, 21000 Novi Sad, Serbia; sebab@uns.ac.rs; 7IFIMUP—Institute of Physics for Advanced Materials, Nanotechnology and Photonics, Department of Physics and Astronomy, Faculty of Sciences, University of Porto, 687 Rua do Campo Alegre, 4169-007 Porto, Portugal; b.postolnyi@gmail.com; 8Department of Nanoelectronics and Surface Modification, Sumy State University, 2 Rymskogo-Korsakova St., 40007 Sumy, Ukraine; 9School of Engineering, London South Bank University, London SE1 0AA, UK; goels@lsbu.ac.uk; 10School of Aerospace, Transport and Manufacturing, Cranfield University, Cranfield MK4 30AL, UK; 11Department of Mechanical Engineering, Shiv Nadar University, Gautam Budh Nagar 201314, India

**Keywords:** additive manufacturing, critical raw materials, CRM, recyclability, powders for additive manufacturing, powder bed fusion

## Abstract

The term “critical raw materials” (CRMs) refers to various metals and nonmetals that are crucial to Europe’s economic progress. Modern technologies enabling effective use and recyclability of CRMs are in critical demand for the EU industries. The use of CRMs, especially in the fields of biomedicine, aerospace, electric vehicles, and energy applications, is almost irreplaceable. Additive manufacturing (also referred to as 3D printing) is one of the key enabling technologies in the field of manufacturing which underpins the Fourth Industrial Revolution. 3D printing not only suppresses waste but also provides an efficient buy-to-fly ratio and possesses the potential to entirely change supply and distribution chains, significantly reducing costs and revolutionizing all logistics. This review provides comprehensive new insights into CRM-containing materials processed by modern additive manufacturing techniques and outlines the potential for increasing the efficiency of CRMs utilization and reducing the dependence on CRMs through wider industrial incorporation of AM and specifics of powder bed AM methods making them prime candidates for such developments.

## 1. Introduction

There is a growing global concern about securing access to metals and minerals needed for developing economic production. The dependence of industrial sectors on scarce materials, in many cases almost entirely dependent on remote sources, represents a threat to the future competitiveness of highly import-dependent industrialized countries such as the European Union (EU) member states, Japan, and the United States. It is also complemented by the additional challenge of sustainable management of all resources starting from raw materials through manufacturing and logistics to waste treatment and end-of-life product recycling. 

Critical raw materials (CRMs) are raw materials of high importance to the global economy. Their supply is at risk, as defined in the current methodology for raw materials assessment published by the European Commission (EC) in 2017 [1]. European legislators were already pointing out the criticality of the raw material for quite a while, indicating that this issue is economic as well as political. In 2007, the EU Council declared the Conclusions on Industrial Policy, requesting the Commission to develop a coherent approach for raw material supply to EU industries. The corresponding approach needed to cover all relevant areas of policy (foreign affairs, trade, environmental, development, and research and innovation policy) and identify appropriate measures for cost-effective, reliable, and environmentally friendly access to and exploitation of natural resources, secondary raw materials, and recyclable waste, especially concerning third-country markets [2]. In response, the first European Raw Materials Initiative was launched by the EC in 2008 to provide a fair and sustainable supply of raw materials from international markets and the EU, while promoting resource efficiency and circular economy [3]. The first CRMs list was released in 2011 and contained 41 candidates, of which 14 CRMs were selected [4] as supercritical. In 2014, the CRMs list was updated, and 20 CRMs were identified out of 54 candidates [5]. A third CRMs list with 26 raw materials and groups of raw materials out of the 78 candidates was released in 2017 [6]. The last CRMs list was released in 2020 and contained 30 elements [7]. Bauxite, lithium, titanium, and strontium were added to the CRMs list for the first time, while helium, critical in 2017, was removed from the list due to a decline in its economic importance. The CRMs list is updated every three years to account for the production, market, and technological developments. A summary of the four CRMs listed above is presented in Figure 1, where elements listed as CRMs in 2011, 2013, 2017, and 2020 are marked in different colors. From the table, the evolution of criticality of each element or material since 2011 is evident. It is worth noting that many other raw materials, even when not classed as critical, are important to the EU economy and are continuously monitored by the EC.

European initiatives were broadened, and in 2011, the EU started a trilateral dialogue with Japan and the United States to promote cooperation in the field of critical materials; identify the main areas of cooperation in collecting raw materials data; and analyze trade, waste recycling, and options for CRM substitution. Representatives of the European Commission (EC); the US Department of Energy (DOE); and Japan’s Ministry of Economy, Trade and Industry (METI) and the New Energy and Industrial Technology Development Organization (NEDO) for Japan decided to meet annually starting in 2011 to discuss CRM issues via a Trilateral Conference.

The growing interest of researchers in solving the problem associated with the supply risk of raw materials is attested to by the increasing number of publications published during the last decade covering topics such as CRM applications, manufacturing, recycling, and life cycle analysis. A simple search through research databases for the keywords, “critical raw materials” returned 333 publications as of February 2021 in the Scopus database alone, with the first publication reported in 1975 [9] and annual publication numbers increasing considerably since 2012. Results also indicate that the publications mostly focus on recycling, substitution, circular economy, and rare earth elements. This search concerns only the publications specifically addressing as keyword “critical raw materials”, while searches for other publications on manufacturing technologies, industrial applications, and disposal of individual elements coupled with the keyword “CRM” would yield even more papers.

The corresponding report on the assessment of the methodology for establishing the EU CRMs list screened 212 communications dealing with critical raw materials, and around 233 organizations were identified as being involved in criticality studies. Among these, 72 organizations developed their methodology, and 58 organizations developed their CRMs lists [10]. In 2018, the Department of the Interior of the United States published the list of 36 critical minerals and elements (including aluminum, arsenic, barite, beryllium, bismuth, cesium, chromium, cobalt, fluorspar, gallium, germanium, graphite, hafnium, helium, indium, lithium, magnesium, manganese, niobium, platinum group metals, potash, the rare earth elements group, rhenium, rubidium, scandium, strontium, tantalum, tellurium, tin, titanium, tungsten, uranium, vanadium, and zirconium) and declared their 100% import reliance on 14 minerals [11,12].

Known approaches to address the problem of CRMs are summarized in Figure 2. They are related to securing the supply chain (through raw materials diplomacy and developing own mining and recycling), extending the lifetime of the products containing CRMs, developing more sustainable production methods for materials containing CRMs, and introducing new CRM-free materials. In absence of having immediate availability of raw materials, novel solutions for improving raw material production, recycling CRMs, reducing CRM consumption, and substituting CRMs move to the top of the agenda [8,13,14].

One of the technologies capable of solving some of the discussed challenges is the additive manufacturing (AM) of metals and nonmetallic materials. AM adds a material layer-by-layer, in contrast to the traditional methods of subtractive manufacturing that remove material from large ingots by turning, drilling, and milling. Unique advantages of AM methods include achieving unprecedented freedom in the shape, significant reduction of waste, and, in many cases, reduction of energy consumption [15,16,17]. Specific processing conditions characteristic of AM allow for developing new materials with unique properties not possible to manufacture by other methods, including bulk metallic glasses [18,19], high-entropy alloys [20,21,22], and different composites [23,24]. Due to these reasons, additive manufacturing was identified as an essential part of the upcoming Fourth Industrial Revolution and, in particular, as an effective and promising method to reduce CRMs use in a wide variety of industrial production processes [8,13]. 

Today, AM technologies are capable of utilizing a variety of different materials. This review focuses on the AM methods capable of working with metallic and ceramic materials most relevant to the CRMs. This review also aims to outline emerging possibilities provided by AM for mitigating critical CRM challenges and to highlight the recent trends in AM of CRMs.

A carefully designed and developed methodology was used to screen the materials presented in this review paper. As such, three separate lines of search were performed in openly available research publications and legislature documents. Corresponding databases used were Google Scholar, Science Direct (Scopus), Springer Link, Wiley Online Library, EU public document and decision databases (Public Register Europa—Europa EU, Documents and publications by the EU—Consilium.europa.eu), and an open Google search.

The first line of search concerned the issues related to CRMs and corresponding future challenges. The material obtained from this search pattern is the basis of Section 1. The second line of search concerned the use of CRMs in additive manufacturing—as individual elements and as parts of alloys. Particular focus was on the precursor material manufacturing methods—AM methodology and material recycling—which have informed the writing of Section 2 and Section 3. The third line of search was partially based on our own databases of research publications and on additional searches on the advantages of AM and future trends in AM development relevant to solving critical issues and future challenges for CRMs.

Since the primary scope for this research is focused on the additive manufacturing of CRMs, the corresponding approach is material- and technology-focused. From this point of view, a full initial database of the publications involved the results of all three searches. Corresponding inclusion criteria were official documents and open scientific publications from peer-reviewed sources. The corresponding initial database was split into three subsets referring to three lines of the search described earlier. One should note that some of the papers are presented in two or even all three subsets.

## 2. Powder Materials Used for Additive Manufacturing

Powder precursor materials are the base for a large family of AM technologies currently used in industry, such as the following:Powder bed fusion (PBF), including selective laser sintering (SLS), selective laser melting (SLM), and electron beam melting (EBM);Nanoparticle jetting (XJET process);Binder jetting printing (BJP);Laser engineered net shaping (LENS).

The requirements for powder precursor materials depend on specific AM technology (see Figure 3). The fundamental requirements for metal and ceramic powders include grain shape (spherical, irregular, granulated), grain size (nano, submicron, or micron powder), composition (pre-alloyed or blended), gas infusions, powder flowability, tendency to oxidize, and sintering/melting conditions, etc. 

Figure 3 illustrates the typical powder grain size distribution required by different AM systems by taking an example of titanium alloy powders produced by gas atomization [25]. EBM uses a nominal particle size distribution between 45 and 106 μm, whilst SLM uses finer powders between 15 and 45 μm. Particle size distribution has a serious impact on the capabilities of the corresponding AM technology. Powders with finer grains allow achieving better control on the layer thickness, which improves print resolution while reducing the as-printed roughness of the components [26]. On the other hand, thicker layers with larger-size grain powder potentially allow faster manufacturing. The presence of finer powder fractions in the distribution allows for higher packing density since small particles help in filling the voids between larger ones, increasing the volume of solid metal produced from the powder layer. Small particles (smaller than 10–15 μm) reduce the flowability of the powder and increase the risks during powder handling. Thus, a trade-off in the particle size distribution is needed to obtain high packing density and good flow properties [27,28].

Powder bed AM technologies, such as SLM, EBM, and LENS predominantly rely on using individual elemental or pre-alloyed powders. During atomization, processing, intermediate handling, and subsequent shipping at air atmosphere, the metal powder can become contaminated, adsorbing gases such as argon from atomization or oxygen, nitrogen, and moisture from the air.

Surface oxide of metal powders (composition, phases/inclusions, and their distribution, thickness, etc.) is connected to the cooling rate and other conditions during atomization, the particle size and secondary dendrite arm spacing, the type of atomization (e.g., water- or gas-atomization or rotating electrode process), and the oxygen availability [29,30]. The undesirable entrapped or adsorbed gases and moisture become the source of pores in manufactured components and can react during the AM process, forming oxide or nitride inclusions and layers at the microstructure boundary surfaces. These oxides result in thin, inherently weak grain boundaries and limit the bonding forces between individual powder particles during AM processing. A powder thermal pretreatment, which involves degassing the powder at an elevated temperature in a vacuum, is one of the possible ways of dealing with the issue. With laser-based AM technologies, it becomes a common addition to the main process, while in EBM it happens inside the machine as one of the essential process stages. Without powder thermal pretreatment, other undesired effects during melting can also occur, including the formation of the “balls” fused randomly to the top of a processed solid layer, severely distorting the process [31]. 

The powder particle size and shape are quite critical. Together with the powder elemental content and level of purity, they play a crucial role in powder selection for AM [32]. It is quite difficult to obtain high-density products with powders that have irregular grains strongly deviating from spherical shape or have large numbers of so-called “satellites” (smaller particles fused with the main grain). In such a case, materials with high porosity and heterogeneity of the microstructure and even anisotropy of properties are commonly obtained. In addition, powder particles with a specific surface have a greater tendency to adsorb gases and humidity from the atmosphere. An important parameter for the overall quality of AM parts is the apparent density of powder before sintering or melting. Though it is not definite, a common “rule of thumb” for EBM suggests that the apparent density of the loose powders should generally be between 50% and 75% for the solidified material. Studies have shown that the control and selection of powder particle shape and size distribution can increase the apparent density of the powder deposited in a layer. Experiments show that the apparent density of thin powder layers increases from 53% to 63% of solid material when adding 30 vol % of fine powder to the coarse one [33]. 

### 2.1. Metal Powders for Additive Manufacturing

Metallic elemental and alloy powders’ grain shape, size distribution, surface morphology and composition, and overall purity are of great importance in the production of good quality and fully dense components [34,35]. This is valid not only for the freshly manufactured powders but for the powders after storage and recycling. Even for the materials with no tendency to easily react with oxygen, the presence of surface oxide can strongly impact the properties of additively manufactured components (e.g., [36,37]). 

The characterization of powders is commonly performed using different analytical techniques including X-ray photoelectron spectroscopy (XPS), Auger electron spectroscopy (AES), scanning/transmission electron microscopy (SEM/TEM), electron backscattered diffraction (EBSD), and X-ray diffraction (XRD). These techniques are limited in terms of either spatial or lateral resolution or chemical information of phases and hence often need to be done together to obtain more meaningful information [38]. Both the characteristics of the metal powder and the type of the AM process determine the properties of the product. Since powder is commonly recycled during AM, the characterization of powder properties is performed not only for the as-received samples but also at regular intervals throughout the manufacturing process.

### 2.2. Production of Metal Powders for Additive Manufacturing

Metal powders can be produced using several methods, some of which are solid-state reduction, milling, electrolysis, chemical processes, and atomization [39]. Atomization so far is the most common route for producing metal powders for AM, dominating the market for powder bed AM. Corresponding atomization technologies are well established. They allow producing powders with different grain sizes in adequate shapes from a variety of metallic materials. These methods are quite cost-competitive and allow for bulk production of powders for both AM and traditional powder metallurgy. The first stage of the overall production chain involves traditional mining and extraction of ore to form a pure or alloyed bulk metal product (ingot, wire, rod). The second stage is powder production itself (atomization process), which is followed by sifting into different fractions, size and shape classification, and validation. For PBF, additional flow tests are commonly added to the validation protocol. 

The specific atomization process can be different depending on the chosen AM technology. Gas and plasma atomization producing particles of quite regular, close to spherical, shape with rather small porosity and high uniformity are the most relevant ones for the powder-based AM (see Figure 4a,b). Annual powder production using water atomization so far well exceeds the volumes produced by gas atomization. However, water atomization results in particles with a highly irregular morphology as the particles solidify faster than their spheroidization time. The resulting powders can contain trapped water and, with some metallic materials, metal hydrides. This renders the water atomization process unsuitable for AM [26].

All atomization processes consist of three main integrated steps: melting, atomization, and solidification. Melting can be accomplished by different techniques such as vacuum induction melting, plasma arc melting, induction drip melting, or direct plasma heating [39]. Though ideal powder grain shape is near-spherical, depending on the method of powder production used, nonspherical particles, joined particles, particles with different intrinsic morphology (e.g., “tear-drop” shapes), and irregularly shaped particles may occur [32]. In some cases, fractions of irregularly shaped grains can be accepted for AM precursor materials if they do not strongly disturb the powder flowability or apparent density. 

#### 2.2.1. Gas Atomization

In gas atomization (GA), the feedstock elemental metal or alloy is melted in a furnace, usually in a vacuum induced melting (VIM) one. The furnace is positioned above the atomization chamber for direct material discharge into the atomizer. In gas atomization, the stream of liquid metal is broken by a high-velocity gas flow (air, nitrogen, argon, or helium) (Figure 4a). Air is commonly used for the atomization of ferrous alloys, and inert gases are used for non-ferrous ones [40,41]. A high solidification rate characteristic for this method results in powders with good material microstructure and quite a homogeneous composition. The particle size distribution can be modulated to a certain extent by adjusting the ratio of gas to melt flow rate. Commercial gas-atomized powders commonly have near-spherical grains with small numbers of attached satellites. The median particle size is in the range of 50 to 300 μm. For a given particle size, cooling rates are about one order of magnitude lower than in water atomization. Some of the powder materials produced by gas atomization are nickel, iron, aluminum, titanium, and cobalt. The characteristic particle size plays a crucial role in the micromorphology, porosity, and gas content of the atomized powders. Pore size and pore presence within powders gradually increase with the increase of average grain size [42]. Although the yield of the fine powder prepared by the GA method is high, such powders are generally characterized by wide particle size distribution and high fraction of hollow powders, which is detrimental to the performance of resultant AM products. Consequently, the yield of the powder with a defined selected grain size fraction after sieving can become significantly lower.

#### 2.2.2. Plasma Atomization

Plasma atomization (PA) has been developed to produce fine, spherical powders. PA utilizes multiple direct-current arc plasma steps to accelerate the atomization gas. In the PA process, metal wires are fed into the apex of the gas plasma flow, where they melt and are atomized in a single step (see Figure 4b). This process offers a unique ability to produce spherical powders of reactive metals with a typical average particle size of 40 μm and the particle size distribution from nanometers to 250 μm [43]. Plasma atomization produces premium-quality spherical powders of reactive and high-melting-point materials such as titanium, nickel, zirconium, molybdenum, niobium, tantalum, tungsten, and their alloys. This process offers the highest purity powders with trueness in the spherical shape of the particles and minimal satellite content. The powder obtained using this method exhibits exceptional flowability and good packing properties [44].

#### 2.2.3. Plasma Rotating Electrode Process

A more specialized method called the plasma rotating electrode process (PREP) makes use of a rotating bar instead of a wire as the source of metal, whereby on entry to the atomization chamber, the bar extremity is melted by plasma torches and solidifies before reaching the encompassing walls of the chamber [45] (see Figure 4c). This process results in powders of high purity, with fairly spherical grains and fine particle sizes (from several nanometers to 100 μm). Titanium and exotic materials can be produced by PREP [26,46]. PREP powder is widely recognized to have very high purity and near-perfect spherical shape. Certain presence of satellites on powder grains not only reduces the fluidity of the powder but also adversely affects the performance of the final products. Present research on Ti-6Al-4V, 316L austenitic stainless steel, and Co-Cr-Mo alloy suggests that it is barely possible to avoid the presence of satellites and joined powder grains during PREP in its present shape [46].

#### 2.2.4. Mechanical Spheroidization of Metal Powders

Certain strategies for improving the powders having irregular shapes after atomization were reported, including mechanical spheroidization of the grains [47]. The flowability of irregularly shaped powders can be significantly improved by tapering sharp edges on the particles through high-speed blending or high-shear milling. Nonetheless, the particles produced by this method are only quasi-spherical, which may limit the applications of such powders. In addition, this method should be used with certain care due to potential mechanical and mechano-chemical effects such as particle surface strengthening and compaction, the formation of oxide and nitride surface films, and changes in material microstructure. Such changes can affect the AM process parameters and the quality of manufactured materials and components.

### 2.3. Metal Powders Processed in Additive Manufacturing

There is a wide range of metallic powders that are already used in AM. The choice of powder depends on the desired properties of the product and employed AM technology. Some of the common metal powders utilized in AM are nickel, steel, aluminum, cobalt–chromium, and titanium alloys. This publication describes issues regarding materials, most of which are not specific to the group of critical materials. However, it should be remembered that not all alloying elements for these materials belong to the critical materials, and in many cases, the CRMs’ share in such alloys is relatively small. However, with growing demand for the additive manufacturing of such alloys, they are widely accepted by the industry when produced by more traditional methods [48]. 

#### 2.3.1. Tungsten Alloys

Tungsten (W) has the highest melting and boiling point among other elements and the lowest thermal expansion coefficient (CTE) among metals [49]. It is mainly produced from wolframite and scheelite; the main producer is China, having about 50% of the world’s reserves.

Among cemented carbides, WC-Co is the main application of tungsten [50]. Studies are reported on the additive manufacturing of WC-12%Co using BJP [51]. The additively manufactured parts passed high-temperature sintering (1485 °C) under a pressure of 1.83 MPa with a resulting density close to the theoretical one—14.1 to 14.2 g/cm^3^.

Another application of tungsten is as an alloying element in high-speed steels for working, cutting, and forming metal components. As an alloying element, W has been used in nickel- and cobalt-based superalloys for aircraft engines and turbine blades because of their high-temperature strength, creep strength, high thermal fatigue resistance, good oxidation resistance, and excellent hot corrosion resistance [50]. Other applications include use in light bulb filaments, electrodes, wires, X-ray and cathode-ray tube components, heat and radiation shielding, and heating elements in furnaces, and these applications account for about 10% of the W market [50].

Tungsten and its alloys can be processed by PBF AM techniques with high-energy-density beams [52,53]. For these alloys, the initial apparent powder density is crucial for the resulting final density of the manufactured components. This means that selection of a proper powder feedstock has a significant impact on the mechanical properties of the manufactured components and should be taken into account during the process parameter optimization.

#### 2.3.2. Chromium and Cobalt Alloys

Despite strategic importance and widespread use, chromium was not included in the CRMs lists released in 2011 and 2017. The main chromium producers are South Africa (producing about two-fifths of the chromite ores and concentrates), together with Kazakhstan (producing one-third of Cr). India, Russia, and Turkey are also substantial producers of Cr. 

Significant chromium demand comes for the production of iron-based alloys. As one of the major alloying elements in stainless steel, Cr content ranges between mass fractions of 10.5% and 30% [13]. Owing to its strong reactivity with oxygen, it provides the ability to passivate the surface by an adherent, insoluble, ultrathin layer that protects the underlying metal against attacks of the corrosive agents, mainly acids and/or chloride-containing environments. Cr is also responsible for surface self-healing in presence of oxygen [13].

Another widespread use of Cr is in surface coatings, such as conversion chromate coatings [54], hard chrome [55,56], and physical vapor deposition PVD CrN-containing coatings [57,58,59,60]. Such coatings are used to improve the resistance of substrates to high temperature, corrosion, and wear. However, electroplated Cr and conversion chromate coatings present health issues and are banned in many applications, with some exceptions for military and aerospace ones. These coatings contain hexavalent Cr, which is recognized to have carcinogenic effects.

In additive manufacturing, Cr is widely used in alloys such as CoCrMo. These alloys are of high demand for specific biomedical implant elements, where high fatigue and wear resistance are of high importance (e.g., knee joints) [61].

The main producer of cobalt worldwide is the Democratic Republic of Congo. Cobalt (Co) is a metal used in several commercial, industrial, and military applications [60]. Co is rarely used as a structural material in its pure form but rather is employed as an alloying element [62].

Stellite is one of the most popular examples of Co-based superalloys. Patented in 1907, originally developed to produce fine cutlery, the stellite alloys have found widespread applications as tool materials for cutting, high-speed machining, etc. Cobalt-based superalloys have higher melting points than nickel-based ones and retain their strength at high temperatures. They also show superior weldability and better hot corrosion and thermal fatigue resistance when compared to nickel-based alloys, making them suitable for use in turbine blades for gas turbines and jet aircraft engines. Stellite can be additively manufactured using direct energy deposition processes [63].

As mentioned above, Co is used in WC-Co cemented carbides that can be processed by BJP [51]. Around 12% of the consumed Co is used for this application, where Co is used as the metal binder due to its excellent wetting, adhesion, and mechanical properties. Additionally, Co is recognized to have genotoxic and cancerogenic activity. 

#### 2.3.3. Natural Graphite and Graphite-Derived Materials

Graphite is a carbon mineral where atoms are arranged in layers with relatively weak bonds between them, granting it high anisotropy in thermal and electrical transport and quite specific mechanical properties [64]. It is used in numerous applications, including electrical machines and vehicles, refractories, foundries, construction industry, and lubricating agents. Natural graphite is mined in three different shapes: vein, flake, and microcrystalline [65]. The bulk producer of graphite is China. Production of synthetic graphite is mainly concentrated in the US, the EU, and Japan, and an increasing trend is forecasted for the synthetic graphite market owing to an increase in demand from the steel and electric battery industries. 

Additive manufacturing using graphite-derived materials (GDMs), such as carbon nanotubes, graphene, graphene oxide, and reduced graphene oxide, is one of the methods intensively developing modern trends [66,67,68,69,70]. It is experimentally shown that the addition of a relatively small (up to 10 vol.%) amount of carbon nanotubes, and especially graphene, can significantly improve the mechanical properties and abrasion resistance of metallic materials (e.g., [71,72,73]). The majority of the experiments were carried out using blends of the main metallic material and fine GDM powders. In these cases, special complicated procedures such as dispersion-based/wet-mixing processes were used to provide a uniform dispersion of GDM through the powder blend [70,74,75,76]. Unfortunately, blending of powders with such dissimilar apparent densities leads to deterioration in GDM distribution uniformity after recycling. However, modern technologies already allow for the effective manufacturing of GDM-coated powders well suited for powder bed AM [77].

#### 2.3.4. Titanium Alloys

Commercial spherical Ti powder production methods include gas atomization (GA), plasma atomization (PA), and the plasma rotating electrode process (PREP). The requirements for particle size distribution (PSD) vary with applications, for example, 20–45 μm for SLM, 10–45 μm for cold spraying, and 45–175 μm for EBM. Most applications require the oxygen content in Ti powder to be less than 0.15 wt.% [40]. Ti-6Al-4V (Ti64) is a widely used α+β alloy known for its enhanced processability and high strength at moderate to high temperatures [78,79]. Aluminum stabilizes the α-phase whereas vanadium stabilizes the β-phase. Due to the high cooling rates during PBF, the β-phase solidifies into primarily α′-martensite microstructures. This leads to embrittlement and decreasing elongation of particles [80,81]. The martensitic phase has the same chemical composition as the β-phase but its crystalline structure is hexagonal and pseudo-compact, resulting in high residual stresses [82]. The α-phase increases hardness and strength, though this also leads to a more brittle sample, whereas the β-phase improves ductility whilst reducing hardness and tensile strength. So far, no comprehensive studies contain a full life cycle analysis of the titanium-based powders used for AM. However, certain conclusions can be drawn from the analysis carried out on the traditional industrially used Ti powders [83].

#### 2.3.5. Zirconium, Niobium, and Tantalum

The promising application of zirconium (Zr) is related to titanium-based alloys. Binary and ternary Ti-based alloys with zirconium, niobium, and tantalum are regarded as the most promising substitution of Ti64 for biomedical applications [84], showing significantly better biocompatibility and having mechanical properties much closer to those of human bones [16,17,85,86,87,88]. Growing demand for prostheses and implants and the ability of additive manufacturing to functionalize them will determine the demand for Zr as an alloying element rather than an individual material.

Zr and Zr alloys are a promising new class of biomaterials. In the past, the main problem of using the powder metallurgy of Zr and Zr alloys was the absence of adequate powder that is possible to use in AM. Patented solutions were not sufficient to introduce this manufacturing technique into the production of zirconium parts. There are many methods for producing Zr metal and Zr powder. The following ones are suitable for powder production: reduction of zirconium dioxide with Ca, Al, Mg, or C; reduction of ZrCl_4_ with Ca, Na, Mg, or Al; reduction of Na_5_Zr_2_F_13_ and K_2_ZrF_6_ with Na, K, or Al; electrolysis of molten mixtures of K_2_ZrF_6_ and electrolytes; and hydrogenation of zirconium sponge or zirconium lump [89]. However, the powders manufactured using these methods are often characterized by elongated shape grains and a high content of impurities. At present, most zirconium products are obtained by foundry methods. New technologies such as direct laser sintering and microwave sintering, used to manufacture high-quality components, require spherical powders with narrow particle size distribution as this affects the packing density and sintering mechanism [90]. Zr, especially in the state of powder, has a very strong activity and strong chemical affinity for oxygen, nitrogen, and hydrogen, so it must be prepared, handled, and processed in tightly controlled technological conditions such as high vacuum and an atmosphere of extra-pure inert gas [91]. Nevertheless, the progress of gas atomization methods already allows for the manufacturing of complex AM-grade powders such as highly biocompatible HEA TiNbTaZrMo ones [92,93]. The powders for the manufacturing of components from pure Zr should also be chemically pure because impurities such as H, O, C, N, and S can cause brittleness. These impurities have a significant influence on metal properties such as tensile strength, hardness, and ductility and increase surface tension during processing. New metal powder processes developed for zirconium synthesis (and the spheroidization) have been developed over the past few years. For example, the South African Nuclear Energy Corporation produces Zr powders for the nuclear industry via a plasma process [94].

#### 2.3.6. Steels and Iron-Based Alloys

Some steels and cast iron alloying elements (chromium, niobium, tungsten, and hafnium) are CRMs or near-CRMs. Thus, iron-based pre-alloyed powders for AM are also the focus of the present paper. Such powders are typically fabricated using advanced powder fabrication techniques such as electrode induction melting gas atomization (EIGA), vacuum induction melting inert gas atomization (VIGA), and plasma atomization. Corresponding powders are high-purity ones and have spherical-shaped grains. In the EIGA process, the metal is melted from an induction-heated rod, from which the liquid metal drops into the atomization nozzle without any contact with the surrounding walls. In the VIGA method, the materials are melted using electromagnetic induction, which delivers heating power into the crucible/material under vacuum or in the inert gas atmosphere without contact with any potentially contaminating material. Once the desired melt homogeneity and chemical composition are achieved, the material is poured into a tundish by crucible tilting. A high-pressure, inert-gas jet atomizes the metal stream flowing from the tundish orifice into the atomization nozzle system. The combination of molten metal and gas jet creates a spray of microdroplets which solidify in the atomization tower and form fine powder with spherical grains [95]. Not all AM techniques are suitable for processing iron-based materials. Specific solidification conditions, including thermal gradients in and around melt pool, and different solidification rates characteristic of AM processes result in different material microstructures. This leads to the differences in phase composition (austenite or martensitic), grain dimension and alignment, and carbide precipitation in the grain boundaries of the additively manufactured steels and high-carbon-content alloys as compared to the materials processed by traditional methods. Nevertheless, proper optimization of the AM processing parameters can lead to materials with superior microstructure and better mechanical properties as compared to traditional manufacturing of the same constituent materials (e.g., [96,97,98,99]).

Iron-based powder grains are typically covered with a relatively homogeneous oxide layer formed by the main element (iron oxide in the case of stainless or tool steels). The thickness of the oxide layer is between 1 and 4 nm, depending on alloy composition, the powder manufacturing method, and powder handling. The rare presence of particulate oxide features with sizes up to 20 nm, rich in oxygen-sensitive elements, was also observed [100]. In many cases, the presence of a thin oxide layer does not impact the quality of manufactured components, but successive powder recycling, especially in the presence of air humidity, can limit the effective lifetime of iron-based powders. Mechanical properties and performance of additively manufactured components can also be improved by post-manufacture heat treatment [97,101]. Other classes of iron-based alloys leading to amorphous materials that have the potential to reduce CRM consumption are discussed in a separate paragraph related to upcoming trends.

#### 2.3.7. Aluminum Alloys

Although aluminum does not belong to critical raw materials currently, a part of alloying elements forming a high number of important aluminum alloys, namely silicon, magnesium, and scandium, are listed as CRMs. Moreover, Al is of high economic importance, and even though it currently has low supply risks, it deserves consideration, already being listed as CRM by the US authorities and as a potential CRM for the EU in the near future. At present, gas atomization (GA) is the main commercial production method for aluminum and its alloy powders [102,103]. Due to the high affinity of aluminum to oxygen, caution should be taken in preventing any possible ignition of the powder or explosion of fine powder fraction suspended in the air. Atomization in air leads to immediate partial oxidation of the liquid material and prevents the liquid metal from transforming into a spherical shape, making the powder unsuitable for additive manufacturing processes. The GA technology for aluminum is a dangerous process and special safety measures are required, which considerably raises the manufacturing costs [104]. The high thermal conductivity of aluminum and its alloys makes them difficult to cast and weld. For powder bed AM technologies, things get worse: aluminum powders are inherently light and have a poor flowability during recoating. They are also highly reflective, creating problems for laser-based AM, and have a high thermal conductivity when compared to other materials [105]. Nevertheless, research on the PBF AM of Al is ongoing. It has been shown that the microstructure of Al-Si (AlSi_7_Mg, AlSi_10_Mg) parts produced by laser methods are characterized by finer grain size in the microstructure as compared to that of cast or wrought parts.

### 2.4. Production of Metal Powders for Additive Manufacturing

Additive manufacturing has already successfully incorporated ceramic materials. According to the form of the precursor, these technologies can generally be divided into slurry-based, powder-based, and bulk-solid-based methods (laminated object manufacturing). The mechanical properties of resulting materials depend significantly on the degree of neck growth between grains, as well as porosity and pore size in the resulting material. Regardless of the specific method, additive manufacturing of ceramics mainly uses materials such as Al_2_O_3_, ZrO_2_, SiO_2_, Y_2_O_3_, TiC, TiN, TiB, AlN, SiC, Si_3_N_4_, WC, Ti_3_SiC_2_, and CaCo_3_. Out of the elements used in the mentioned ceramic materials, only silicon, cobalt, and tungsten are on the CRMs list, with zirconium and aluminum expected to be on the CRMs list in the near future. However, ceramic and ceramic-containing materials have the potential for substituting some of the CRM-dependent ones and thus deserve corresponding analysis. 

In solid-phase reaction synthesis of ceramic powders, there are three types of chemical reactions: oxidation or reduction of a solid, thermal decomposition of a solid, and solid-state reaction between two types of solid. With liquid-phase synthesis of ceramic powders, there are five different methods: drying of a liquid, precipitation, sol–gel synthesis, hydrothermal synthesis, and reactions of a liquid metal melt with gas to give a solid ceramic. There are three operational principles for precipitation: temperature change, evaporation, and chemical reaction. These methods are generally broken into three categories, namely solid-phase reactant, liquid-phase reactant, and gas-phase reactant synthesis, and gas-phase reactant synthesis is essentially a precipitation method; however, the solid precipitated is of nanometer size and can be organized into a gel network or sol particle depending on conditions. Hydrothermal synthesis methods use high pressure to make a specific solid phase insoluble. Gas-phase ceramic powder synthesis methods include evaporation–condensation and chemical reactions in the gas phase. These gas-phase reactions include thermal decomposition, oxidation, or reduction, as well as chemical combination reactions [106]. The most common is the use of AM for Al_2_O_3_ and ZrO_2_ [107]. It is known that when using free sintering or pressure sintering methods, the highest relative density values and thus the best mechanical properties are obtained for very fine powders, preferably sub-micrometer ones (Figure 5a). Commercial powders are usually available in the form of weak agglomerates or granules prepared from very fine powders (Figure 5b). Isometric shape particles and granules of ceramic powders are preferred in free and pressure sintering processes because of the better formation and consolidation of the grains. Many multicomponent nanosized ceramic powders have been prepared using an aqueous sol–gel method.

In industrial production, the granulation methods of ceramic powders mainly include dry roller granulation, cold isostatic pressing, and spray granulation. In the case of free and pressure sintering, small amounts of additives, e.g., MgO to Al_2_O_3_, Y_2_O_3_ to ZrO_2_, or carbon for SiC sintering, are introduced into the powders. These additives limit grain growth, stabilize selected phases that are desired to be kept, improve the stoichiometry of the product, and facilitate sintering by lowering the sintering temperature. 

A large part of the research conducted in the AM of ceramics field is based on powders with a larger size of 40–100 microns. These powders are characterized by lower relative density, and this determines the lower strength of the sintered contacts (necks) after sintering (using AM methods), which is the basic problem of using the indirect AM manufacturing of ceramics. For this reason, in order to increase the density of additively manufactured ceramic products, finer-size powder is fed into the process of granulation or functionalization of their surface in order to improve the flowability and sintering performance of these powders. High values of particle spheroidization and fractional composition homogeneity are achieved after plasma treatment. A comparative study of thermal barrier coatings based on yttria-stabilized zirconium oxide powder demonstrated that deposited coating thickness, powder dispersion degree, and material efficiency of plasma-spheroidized powder are comparable to those of a high-quality commercial powder [108].

### 2.5. Ceramic Powders for Direct Additive Manufacturing

The direct additive manufacturing of ceramic components is still at an early phase of development, although it was attempted by Lakshminarayan et al. [109] in the 1990s. For some AM processes producing ceramic parts, cracks are still the most critical flaws that compromise the mechanical strength. During single-step processes, i.e., direct energy deposition and single-step PBF processes, thermal cracks are generally caused by thermal shocks introduced by the laser beam heating [110]. The direct AM process is very challenging due to the ceramic material properties, such as high melting temperature, high melt viscosity, and poor thermal shock resistance. Sources such as focused lasers and electron and infrared beams are used as heating–sintering tools. The process of heating allows the powder to take the shape of the intended object. This greatly improves the productivity of additively manufactured ceramic components because the time-intensive debinding and sintering phases characteristic of indirect methods are not necessary. The use of granulate composed of micrometric yttria-stabilized zirconia with sub-micrometric alumina improved the homogeneity of the microstructure. In some cases, thermal post-processing can improve the mechanical properties of the resulting material. For example, it allows the amorphous alumina in corresponding ceramic parts to crystallize [111]. An important phenomenon that should be taken into account during ceramic powder consolidation by direct AM methods is the formation of the glassy phase, which can affect the fragility of products.

### 2.6. Ceramic Powders for Additive Manufacturing of Metal-Ceramic Composites

One of the promising applications of ceramic powders in AM is using them together with metallic ones for producing metal–ceramic composites (MCCs) in AM processes initially developed entirely for metal precursors. Experiments carried out using different AM technologies [112,113,114,115] indicate that this method allows improvement of mechanical properties and the abrasion resistance of the basic alloys. Experiments were carried out using powder blends, in which the ceramic phase was a very fine powder, and using different technologies providing agglomerated grains containing both ceramic and metal powders (e.g., [116]). Different mechanisms responsible for property improvement were suggested, including the ability of sub-micrometer ceramic inclusions to act as dislocation traps. The resulting microstructure strongly depends on the melting temperature of the ceramic and the temperatures reached in the melting pool and on the wettability of the ceramics in the molten metallic material. At the current stages of research, it is not possible to forecast which combination of materials in MCC AM will be successful in producing materials with superior properties. However, this line of development definitely has potential in relation to sparing CRMs in industrial applications.

### 2.7. Ceramic Powders for Slurry-Based Methods

Slurry-based ceramic 3D printing technologies generally involve fine ceramic particles dispersed in liquid or binder in the form of relatively low viscosity inks or viscous pastes. The slurry content can be additively manufactured by photopolymerization, inkjet printing, or extrusion [117]. All slurry methods are commonly multistep ones, initially producing nondense semifinished parts that are commonly called “green bodies”, followed by debinding and firing processes yielding final components. 

Binder jetting is an additive manufacturing process in which a liquid bonding agent is selectively deposited to join powder materials [118]. Currently, the density of the ceramic parts made by binder jetting is rather low, and their mechanical properties are far from adequate. The main reason comes from the low sinterability of current powder feedstock due to large particle size (10–100 μm) and the inability to deposit a smooth layer of the precursor. The coarse powder exhibits good flowability, and the fine powder that can provide better sintering has poor flowability [119]. Many studies have reported that the quality of parts using binder jetting is significantly different when coarse powders are used. Studies have shown that the accuracy and strength of ceramic parts are closely related to powder and binder choice, printing parameters, equipment, and post-treatment. Studies have focused on the optimization of binder jetting employing multimodal filler particles for improving the strength and performance of binder-jetted parts [120]. One of the solutions to improve the compaction of the material is the use of nanopowders. Smaller particles as densifiers occupy the intergranular pores in the powder and improve the density of green-printed parts, but the applied nanosuspension can quickly clog the jetting nozzles [121]. The shape of ceramic powders mainly affects the flowability of slurry, the tap density, the powder bed (packing) density, the pore structure of the green body, and the contact mode between the particles. Generally speaking, spherical particles have better flowability in the slurry and higher tap densities than irregular ones. However, during the printing process, the powders will be spread by the roller, which means that the powders will not be compacted; thus, the contribution of spherical morphology to packing density will be reduced. In contrast, irregular powders have a relatively high packing density [122]. Suwanprateeb et al. [123] reported that irregular hydroxyapatite has a higher packing density than spherically shaped powder. This is because the spherical particles undergo a low uniaxial pressure, and their good flowability causes the particles to roll towards each other. Although the particles are rearranged and slipped, they are still in point contact and, thus, cannot effectively reduce the pore volume. For irregular particles, after being rearranged and slipped, the larger internal friction causes them to combine and become compact, while the point contact between some of the particles becomes surface contact, which can effectively reduce the pore volume. Therefore, the irregular powders will result in a higher green density than the spherically shaped ones. The green body density is usually positively correlated with green strength. This higher green strength improves the handling characteristic of the as-fabricated green body. The original morphology of as-purchased hydroxyapatite powders prior to preparation commonly exhibits agglomerates of needle-like crystals [123].

A new powder surface modification method, i.e., the particle coating sol–gel process, was used to synthesize the amorphous phase material and was applied to increase powder sinterability and part strength. Specifically, coarse crystalline alumina particles (70 and 10 mm on average) were coated with amorphous alumina, in which the microsized core was designed to provide high flowability and the amorphous shell to promote sintering due to its high activity [124]. The coarse crystalline core can help to maintain the high flowability, and the amorphous shell can promote sintering due to its high activity [124].

### 2.8. Ceramic Powders for Porous Bone Implants

While research on ceramic scaffolds for bone regeneration has progressed rapidly, the clinical outcome of these synthetic bone implants remains limited, especially for major load-bearing applications. These scaffolds should not only provide adequate mechanical support but also possess sufficient porosity to facilitate nutrient/metabolite transportation and bone tissue ingrowth [125]. At the same time, ceramic implant-scaffolds have a great potential for replacing metallic ones due to their advanced biocompatibility, reducing the dependence on certain CRMs traditionally used in metallic implants in future orthopedics.

One of the additive manufacturing techniques, direct ink writing (DIW), also known as robocasting, has attracted considerable attention in bone tissue engineering. In the robocasting fabrication method, a filament or ink is extruded through a nozzle in a layer-wise fashion and ultimately forms a 3-D mesh structure with interpenetrating struts. After the initial layer is created, the X–Y stage is incremented in the Z-direction and another layer is deposited. This process is repeated until the desired scaffold structure is created. While robocasting can fabricate regular and controllable patterns in the X–Y plane, its ability to maintain high precision with sophisticated structures in the Z-direction is restricted due to depositing ceramic struts on top of one another [126]. This technique has been used to fabricate scaffolds with a wide variety of ceramic materials such as bioactive glass [127], hydroxyapatite (HA), calcium phosphates [128], calcium silicate (CSi), and Sr-HT Gahnite [129], as well as other composite materials, exhibiting significant potential. Polylactide or polycaprolactone scaffolds with pore sizes ranging between 200 and 500 μm and hydroxyapatite content of up to 70 wt %, as well as scaffolds containing bioactive glasses, were also 3D-printed [130,131].

Ceramic scaffolds and implants for osteogenesis are based mainly on hydroxyapatite since this is the inorganic component of bone. The usual fabrication technique for ceramic implants is the sintering of the ceramic powder at high temperatures.

Porosity control in ceramic additive manufacturing is quite challenging. One should distinguish between scaffold porosity and material porosity. Scaffold porosity mainly relates to the ratio of the solid material to the free space in the manufactured scaffold or “porous” implant section. This property is strongly related to the part design and ability of the material and chosen AM method to produce the part without deviating from the designed shape. As a rule of thumb, in AM structures, it is very hard to design holes smaller than 5 times the average size of the powder grains. Pores (micropores) in resulting solid materials (e.g., struts in the porous-by-design lattice scaffolds) are mainly related to the material, AM technology, and process parameters. These micropores commonly have different shapes and sizes, and their distribution is not uniform. The micropore morphology can be partly influenced by controlling the size distribution and morphology of the precursor powder. The porosity of materials can also be controlled by an appropriate selection of sintering conditions (time, temperature, pressure, atmosphere) [132,133]. For example, hydroxyapatite samples additively manufactured from milled powders are significantly stronger than samples manufactured from spray-dried powders. This is a combination of the specifics of the manufacturing and the difference in morphology of the prepared powders. In the case of milled powders, these factors induce better packing and rearrangement in the green state and improve densification and pore characteristics in the sintered state. Although the spray drying technique of powder preparation is more convenient and faster, the grinding route is preferable when the greater strength of fabricated components is considered [123]. Another ceramic material for bone implants is bioglass (materials with different compositions of SiO_2_, CaO, Na_2_O, and P_2_O_5_) [134].

### 2.9. Powder Handling Safety Issues

Safety precautions in handling CRM-containing powders used in AM are always mandatory. Many of the CRM-containing materials are listed as “dangerous” in quite different ways, so studying the safety precautions and safety data sheets related to the involved chemical elements and materials is advised. Handling with care, i.e., avoiding spillage and anything promoting contaminating the air with fine material powders, is always advised. Fine particles can cause severe dysfunctions, skin problems, lung diseases, or cancer upon exposure or inhalation. Prolonged exposure to some of the metals was linked to the onset of Alzheimer’s disease [135]. Special powder-safe respirators should be used to prevent small particles from reaching the bronchus and lungs, and powder-free gloves should protect the hands. 

In addition to potential health risks, metal powders are combustible and flammable; when aerated, they present a risk for explosion. Facilities where metallic powders are kept or handled should have proper protection from electrostatic and electrical sparks (including nonstatic flooring, special clothing and shoes for the staff, grounding wires, and special vacuum cleaners, as the majority of domestic vacuum cleaners have spark-producing electric motors). Additionally, only specialized fire extinguishers rated for combusting metals should be used in such facilities. 

At the same time, with correctly deployed preventive measures and proper handling protocol implementation in corresponding AM facilities, levels of danger are no higher than those found in many common industrial facilities.

## 3. Additive Manufacturing Processes

The main consolidation mechanisms of AM technologies are partial melting, full melting, and solid-state sintering, which might act together, and it is not always obvious what consolidation mechanism is dominating [136].

The role of additive manufacturing is going to increase as the world enters the Fourth Industrial Revolution. The following key advantages of the AM methods will define their fundamental role in future manufacturing: Additive manufacturing is capable of extremely high flexibility in producing small and medium series of complex parts, complemented by seamless switching to the manufacturing of the parts with a completely different design. Manufacturing capacity with the AM methods is highly scalable, which would be beneficial to both industrial giants and small and medium enterprises (SMEs). AM methods allow not only a higher flexibility to achieve any desirable shape but also cost- and time-effective functionalization and individualization of the parts. The digital nature of the design for AM allows for reducing cost and time for component modification which reduces the need for inventory as they can be manufactured on demand using the library of digital files. With the development of the “service points” for additive manufacturing across the world, there is a potential for a significant reduction in transportation costs for raw materials and, especially, manufactured components. Recycling of the precursor powders can be almost completely performed at the manufacturing sites. The high degree of recyclability of the powders, along with other aspects of material and energy saving, allows for significantly decreasing the environmental impacts of industrial production. Along with the possibility of reducing the amounts of CRMs per component, and with newly developed materials with the ability to avoid using them, additive manufacturing will be a major contributing factor in solving the CRMs problem and reaching the goals set by EU Commission.

### 3.1. Industrial Additive Manufacturing for CRM-Containing Materials

Perez et al. [137] presented the general AM standards related to terminology, data formats, design rules, and qualification guidance and generally outlined different additive manufacturing technologies. A comprehensive overview of AM processes and standards with emphasis on materials, processing, and testing methods is given by Riipinen et al. [138]. The well-established additive manufacturing processes are classified according to ISO/ASTM 52900-2017 into single-step and multistep AM processes [118]. These are further divided into processes where AM is done as a fusion of similar materials or as adhesion of dissimilar materials. In both groups, processes are divided according to used material classes into metallics, polymers, ceramics, and composites, and secondary processing, such as sintering and infiltration, is also mentioned. Figure 6 presents the most mature and widely used AM processes such as vat photopolymerization (VPP), binder jetting (BJ), powder bed fusion (PBF), material jetting (MJ), direct energy deposition (DED), sheet lamination (SL), and material extrusion (ME), together with materials that can be used in a particular process. A more complete interactive map of additive manufacturing processes, which also includes materials such as cement, hydrogels, and bioinks, with a directory of more than 800 companies manufacturing AM hardware can be accessed at 3dprintingmedia.net [139]. 

Today, the most promising AM technologies from the CRM point of view are powder bed fusion (PBF) and direct energy deposition (DED), which are also the most widely used in the industry [141]. These are followed by binder jetting and material extrusion (ME) [142,143]. The less industrially known AM technologies, which use ultrasonic, friction, and friction stir welding, together with thermal spraying (e.g., cold spraying), also present high potential [144,145,146]. Their main advantages compared to PBF and DED are in the possibilities to join dissimilar materials, better energy efficiency, smaller heat input, a protective chamber or atmosphere generally not being needed, and a promising buy-to-fly ratio.

Within PBF AM, one can distinguish between SLM and EBM technologies. Both of them use powder precursors, and the manufacturing is performed in a closed chamber in an inert atmosphere (SLM) or high vacuum (EBM). In both PBF AM technologies, selective melting of desired parts of layer of powder by intense beam, with a nonmelted powder enabling partial support of manufactured components and acting as a heat insulator, makes the product. The process continues by adding and melting consecutive layers of powder. 

Laser-based PBF AM methods allow for industrial manufacturing of functional parts and tools with complex shapes from metals (stainless steels, tool steels, Co-Cr alloys, Ni-based superalloys, Ti alloys, Al alloys) and ceramic powders. The component surfaces have excellent to moderate finish in an as-manufactured state, good functional properties, and can be micro- and nanostructured. The products are usually made from one material, which enables high recyclability of the powder. Powder preheating is done using infrared heaters or beams of lower intensity, and laser beam deflection is done using mirrors. 

Electron beam melting generally enables working beams of higher intensity and higher (up to 60 cc/h) material deposition rate but is generally limited only to the electrically conductive powders. This defines the quite limited selection of materials available for industrial EBM manufacturing, including Ti (grades 1, 2, 23, and 5), nickel-base alloys, aluminum alloys (Al-Si), stainless steel (316L, M300, 17-4 PH), and CoCrMo [44]. It also enables the production of functional parts of complex geometry, but its resolution is lower and the surface roughness of components is higher than those provided by the laser-based methods due to the larger average grain sizes of precursor powders. The process is carried out in a deep vacuum, which enables close to 95% energy efficiency, which is 5–10 times higher than that for laser-based PBF. The vacuum is also perfect for processing reactive metals like Ti and Al, maintaining the chemical composition, and reducing the heat loss to the environment. The additional advantage is due to the powder bed preheating. Being an essential prerequisite for the conductive powder melting (partially sintering it and preventing the formation of clouds of charged particles in the working chamber), it efficiently removes moisture and gases adsorbed on the powder surfaces. EBM-manufactured parts also exhibit significantly lower internal stress, as the process is carried out at elevated temperatures, continuously annealing the produced material. 

Corresponding disadvantages of the electron beam PBF as compared to the laser-based ones are smaller available working volumes (and thus maximum component dimensions), complications of the semi-sintered powder recovery from holes and crevices, higher surface roughness of components in as-manufactured state, and longer times for reloading the machines. Additionally, laser-based AM technologies are already extensively incorporated into the hybrid manufacturing chains using precise machining after AM, and they can provide higher technology readiness levels of as-manufactured parts. At the same time, it should be noted that both laser- and electron-beam-based PBF AM technologies are continuously developing and are slowly overcoming their current limitations. 

SLM, another PBF AM representative, enables the production of complex parts with high technology readiness levels (TLRs) for aerospace (TLRs 5–7), tooling (TLR 9), automotive (TLRs 4–5), and medical industries (TLRs 9–10). Typical products for the aerospace industry are fuel injection components, structural elements, and turbine blades. In the tooling industry, inserts are made, usually with complex internal cooling channels that enable efficient cooling/heating and prolongation of tool life. In the automotive industry, different structural and functional components are successfully manufactured in low series. In the medical industry, medical instruments, artificial hip joints, different implants for reconstructive surgery, and dental crowns and copings are produced [143,144,147,148,149,150]. 

Another powder-using AM technology, DED, is less widespread for metal AM as compared to PBF ones due to lower accuracy and requirement of post-processing. However, it enables higher deposition rates, production of bigger parts without limitation to small chamber size, component reparations, and production of functionally graded materials and multimaterial components. Corresponding equipment includes a feeding “nozzle” mounted on a multiaxis arm or robotic system with an external rotating table and protecting chamber in case of manufacturing using reactive metals. The material in powder or wire form is supplied through the nozzle and is melted by the electron or laser beam together with the surface of the product. The wire is a cheaper precursor material; it enables higher deposition rates, but such systems have lower accuracy compared to powder-based ones. Materials in powder form are more expensive, and the atomization process of powder production is less energy efficient (11–59%). In contrast to wire-based systems, powder ones can return into processing from 20% to 90–98% of precursor directly or after sifting [135,142,143,144,145]. However, the wire is cheaper, more widely available in larger quantities, and demands less stringent safety precautions in handling compared to metal powder. Wire, with its much lower surface-to-volume ratio, is less prone to absorbing moisture, nitrogen, oxygen, and other undesired elements from the atmosphere, and thus it affects the deposition process much less and provides materials with lower numbers of residual pores as compared to powder-based technologies. An interesting new technology that will soon be industrially available is the so-called “Joule Printing”. It is quite similar to wire-based DED AM technology but uses resistive heating (as in the welding process) instead of electron or laser beams. Its advantages are lower energy consumption and low heat input (1.4–1.6 Wh/cc), which only transforms the precursor material into a mushy state and avoids the formation of the melt pool. The process enables producing near-net-shape structures with high deposition speed (1000 cc/h), which is 2–10 times faster than DED-powder, and similar rate to DED-Arc, while the resolution is comparable to that of DED-powder [151]. Manufacturing can be done in demanding environments, with high material efficiency, using commercially available welding wires, while the output material properties are close to wrought/cast metal ones [151]. 

In general, DED enables the production of less complex parts as compared to PBF. DED parts usually need post-processing (e.g., heat treatment), mechanical treatment, and machining to obtain the desired shape and mechanical properties. DED is typically used for repair of worn components, modification of tooling for reuse, rapid prototyping of bigger parts, and direct manufacturing of large components. DED processes are generally classified as electron beam additive manufacturing (EBAM), laser metal deposition (LMD wire/powder), and wire and arc additive manufacturing (WAAM) [151,152,153,154].

EBAM uses precursor material in the wire form. It produces bigger near-net-shaped parts of the highest quality inside the vacuum chamber. Parts are made with high deposition rates ranging from 3 to 11 kg/h. In the process, a variety of materials can be used: titanium and titanium alloys, Inconel, nickel alloys, stainless steel, aluminum alloys, cobalt alloys, zircalloy, tantalum, tungsten, niobium, and molybdenum [155]. During the EBAM and post-machining, high material efficiency is achieved overall [154].

LMD uses the material in a wire or powder form, which is fed through the nozzle. The fabrication can be done in a local protective atmosphere or an inert gas chamber. Deposition layer thickness varies between 0.1 and ~3 mm, deepening if a high-resolution printing or fast manufacturing rate is targeted. Deposition rates for wire LMD are much higher than for SLM or powder LMD. A vast variety of different materials and alloys can be used for processing. Good metallurgical bonding with a low dilution level can be achieved with low impact on base material properties [154].

WAAM uses an electrical arc from power sources typical for welding such as gas metal arc (GMA), tungsten inert gas (TIG), plasma (PL), or their combination together with the precursor material in a wire form. The process achieves high deposition rates from 1 to 10 kg/h, with moderate surface finish and low to medium part complexities. Major benefits are lower equipment and precursor material cost, a big variety of precursor materials, and high material efficiency [152,153]. This makes the process extremely well suited for the manufacturing of large parts with up to medium complexity, at a much lower price compared to other AM processes. The main drawback is potentially higher residual stresses, distortions, and coarse grain microstructure as the consequence of higher temperature gradients. Additional post-processing with mechanical treatment, heat treatment, and/or machining may be needed to obtain industry-acceptable products [96,97,152,153,156,157,158]. 

Table 1 presents a comparison of the most common PBF and DED additive manufacturing processes in terms of processing conditions, component complexity and qualities, deposition rates, and multimaterial processing capability. The lack of multimaterial processing capability stated for SLM and EBM is related to the industrial-grade processing, although experimental confirmations of its feasibility have been demonstrated.

### 3.2. Recycling of Metal Powders for Additive Manufacturing

Powder recyclability is a crucial parameter of powder lifecycle and overall manufacturing efficiency. In metal-based processes (PBF, LENS) the microstructure of the virgin (not recycled) powder has a certain tendency to change due to repetitive reuse and recycling [14,166,167]. The flowability and powder morphology can change because of thermal cycling during processing and mechanical impacts during layer deposition, powder recovery, and sifting. Increased temperatures of the powder in the AM process can force surfaces of powder to react with ambient atmospheric gases or their residual content in the protection volume. An increase in oxygen content in the powder often results in reducing the mechanical performance of the printed metal parts [14,166]. Most often, powders are passed through a sieve before being used again. That may cause particle deformation and breakage of the grains that form joining necks. Recycled powders show a minor decrease in the amount of fine (<10 μm) particles, a slight increase in average particle diameter, and a slightly wider grain size distribution [14,168,169]. Sieving also can remove some of the satellites, leading in some cases to better flowability of the recycled powders. 

Additional contamination by impurities coming from the sieves and vessels, foreign bodies, or interstitial elements may be introduced to the powder as a result of handling during pre- or post-processing stages [26]. It is clear that the amount of powder belonging to each separate virgin batch constantly decreases as part of it forms solid components. At some point, the volume of the powder becomes lower than the minimum demanded by the AM machine. Three strategies are commonly used: topping up the reused powder with virgin one, saving small batches of recycled powders and mixing them for further manufacturing, and using recycled powder from a virgin batch without topping or mixing at all. The third option is preferable for manufacturing parts destined for critical applications. However, there is no consensus on which of the first two options should be preferred and in which cases. Nevertheless, it is agreed that each recycled powder should have its “passport” stating the date of virgin batch purchase and its initial elemental content, history of the builds using it, number of recycling procedures, and results of regular powder analysis. It is also clear that for increasing powder lifetime, the recycling process and storage conditions for the powder, including humidity and temperature control and specialized ventilation with filtering of the incoming air control in the operating rooms, should be strictly regulated. 

An interesting approach of using beam-based technologies for material recycling is also starting to develop. Recently, some authors have investigated the feasibility of turning recycled powders that are not fit for further use due, for example, to some excessive agglomeration, into metal bars [170]. A powder bed machine manufactures thin-walled cylinders with the powder enclosed in them. This does not require high purity of the precursor and is not depending on the high spatial resolution. This is followed by hot isostatic pressing (HIP) treatment of the cylinders, resulting in solid metal bars. Studies have shown that it is quite realistic when using EBM, as the processing happens in a vacuum, and powder inside the sealed cylinders does not have any encapsulated gases. This approach is inspired by the recycling pathway using laser- or electron-beam-based equipment with simplified control of the beam used just for melting scrap material into ingots [171,172,173], but it has the advantage of conducting essentially full recycling at the manufacturing site without any need for costly and hazardous transportation of powders to a specialized recycling site. 

## 4. Modern and Future Trends in Additive Manufacturing of CRM-Based Materials

Table 2 outlines the modern status of additive manufacturing of CRMs and CRM-containing materials. Further progress in solving CRM-related challenges belongs to the developing trends in AM and its deep integration with other processing modalities. Several AM-related possibilities to reduce and optimize the use of CRMs and CRM-based materials are discussed in this review: Use of hybrid manufacturing [138,146,174];Production of multimaterial components [175,176];Production of functionally graded materials (FGMs) [153,165,177,178];Repairing and remanufacturing using additive manufacturing [143,178,179,180,181,182].

This review also provides a brief overview of laser shock peening, one of the potential industrial post-processing routes allowing for reducing residual stresses in additively manufactured components and increasing their value as CRM-reducing and CRM-sparing manufacturing routes.

### 4.1. Hybrid Manufacturing Technologies

Hybrid manufacturing is commonly described as the combination of additive manufacturing and subtractive manufacturing in a single machine or a set of closely linked machines in a single production line [222,223]. Manufacturing of the component in such a case could start from a billet or a plate and be followed by pre-machining, manufacturing, and post-machining stages, where AM technology is used in one or more of these stages. Each of the stages most efficiently uses its best advantages, thus producing a final product with high complexity, high precision, and best surface finish. In hybrid manufacturing mode, parts with precise complex geometry, multimaterial components, parts with conformal cooling channels, multimaterial tools, etc., can be produced with high efficiency and optimized use of precursor materials. Lightweight structures, turbine blades, different housings, structural elements, and multimaterial components for the aerospace and automotive industries can be manufactured by optimizing the use of CRMs and increasing the buy-to-fly ratio [224]. Different medical tools, fixation elements, and implants with patient-specific and functionally optimized geometries and elements for optimal tissue ingrowth, biocompatible and bacteria-hostile surfaces, and suitable mechanical properties can be efficiently produced by AM, while their precision elements (threaded holes, sliding balls with mirror surface finish, etc.) can be effectively completed by traditional machining. Many of the post-processing stages such as additional machining, polishing, heat treatment, and surface modifications that today are separated from AM technology can be included in the hybrid manufacturing systems.

Reduced consumption of CRMs can also be obtained using repair welding and repair additive manufacturing of worn out or damaged products, so the need for new parts is significantly reduced [225,226]. This option is increasingly utilized in the industrial tooling sector, where an improved repair welding (cladding) technology and proper selection of materials can lead to the restoration of the tool several times. This approach not only reduces the use of CRMs but also saves other resources and reduces the waste and greenhouse gas footprint of the manufacturing.

### 4.2. In Situ Alloying

CRMs used in aerospace and biomedical industries are the main constituents of composition-wise and microstructurally complicated alloys with mechanical, physical, and thermal properties adjusted to the service conditions of the critical components. Today, pre-alloyed powders are the main precursor for additive manufacturing of CRM-containing materials. However, the low availability and high cost of quality-produced atomized spherical pre-alloyed powders may be a vulnerable point in the production chain of the critical components. In addition, not all desired materials can be alloyed effectively in stationary conditions, further limiting the possibilities of additive manufacturing of CRM-containing materials. Despite numerous advantages of AM, it hampers the development of technology implementation as a preferential production route for CRM-containing components [96,227,228,229,230,231]. In situ alloying of blended/elemental powders during the AM process enables overcoming of this obstacle [232].

**Titanium-based alloys** belong to the most popular CRM-based ones used in aerospace, automotive, and biomedical applications with the domination of precursor materials in the form of pre-alloyed spherical powders [183,233]. Certain attempts to perform PBF AM of microstructurally complicated Ti-based materials via successful in situ alloying of elemental powder blends were performed during the last years and reported in the literature. 

Vrancken et al. [234] studied SLM in situ alloying of Ti64 and 10% Mo powders. Molybdenum powder with 5–10 μm sized particles was used. SLM-produced alloys have a good combination of high strength (919 MPa), excellent ductility (20.1%), and low Young’s modulus (73 GPa). 

Dzogbewu et al. [235] reported SLM manufacturing of Ti15Mo alloy for biomedical applications using Ti and Mo elemental powders as raw materials. Various blend compositions and various scanning strategies were studied, and the obtained as-printed materials were characterized to optimize the process parameters. It was concluded that although achieving the final product with good homogeneity remains a challenge, in situ alloying involving beam-based AM like SLM has a high potential for developing new materials. 

Yadroitsev et al. [236] reported on the trials to alloy Ti15Mo and Ti64 with Cu and Mo introduced as elemental powders. The effects of the process parameters, i.e., energy input and scanning strategy, were studied. The viability of the in situ alloying assisted SLM as a production route was confirmed. 

Fischer et al. [237] investigated and reported the microstructure and mechanical properties of Ti-26Nb alloy synthesized by powder bed SLM from a mixture of Ti and Nb elemental powders.

Surmeneva et al. [220] reported on the in situ alloying of binary Ti10at%Nb by EBM of elemental powders. Despite quite unfavorable grain size distribution and shape of used Nb powder, it was possible to achieve test samples of reasonable quality. At the same time, the uneven distribution of Nb fraction throughout the material led to the local gradients in the Ti-Nb mixing ratios. Though unfavorable for industrial process, this allowed for studying the microstructure of EBM-processed Ti-Nb materials with different elemental content. It was concluded that further process parameter optimization and agglomerated powders in adjusted mixing ratios are essential for effective in situ alloying and production of porosity-free Ti-Nb alloys. 

**Nickel and iron-based alloys** are classified as high-performance structural materials and are widely used in aggressive environments and at elevated temperatures. The urgent need for components having compositional and functional gradient combined with high geometrical complexity causes a growing interest in the implementation of AM manufacturing routes for these alloys [238].

Li et al. [238] studied and reported on in situ alloying assisted SLM synthesis and characterization of Fe-Cr-Ni alloy from a pre-mixed blend of elemental metal powders. The authors investigated precursor materials (blending ratios and particle sizes) and manufacturing parameters (heat input and scanning strategy). Phase compositions and microstructural formation were thermodynamically calculated and predicted and compared with the obtained experimental results. The corrosion resistance of the synthesized functionally graded material was examined, and the applicability of the in situ assisted thermal AM-SLM route was confirmed. 

Li et al. [239,240] synthesized and investigated a novel heterogeneous material, alloying Ti64 and SS316 with multimetallic fillers. The authors concluded that the studied material is promising for critical spacecraft components, which require lightweight, high strength to weight ratio, and corrosion-resistant materials. 

Shah et al. [241] synthesized an Inconel stainless-steel-based functionally graded material with strong corrosion resistance at high temperatures. This material should be used as a critical raw material in light-water reactors subjected to a large variety of high temperatures, pressures, and stresses [242].

**High-entropy alloys (HEAs**) are a novel promising class of materials, in which the formation of a single-phase solid solution is thermodynamically preferable over the formation of intermetallic compounds [243]. Refractory metals containing HEAs are usually composed of body-centered cubic (BCC) solid solutions [244] and have a high potential to substitute presently used critical raw materials due to their high-temperature mechanical strength [244]. Sometimes, full or partial substitution of refractory elements such as W, Ta, and Mo with transition metals [245,246] or Al [247] is performed to decrease the specific weight of the alloy and to improve corrosion resistance. Although this substitution is useful for achieving the mentioned aims, it may result in a poor mixing of the raw constituents, which causes low homogeneity of the finally obtained material. Furthermore, since the conventional way of producing HEAs is vacuum arc melting [248,249,250,251], the main problem arises due to the difference in melting points and vapor pressures of the alloying elements at high temperatures. Additionally, oxidation resistance of the alloying refractory elements requires an extremely high operating vacuum, at which low-melting constituents usually evaporate. The problems mentioned above bring PBF AM technologies into focus as the most promising manufacturing technology for this class of materials. However, some of them contain CRMs, but there is a high potential of an overall reduction of their use when HEAs substitute more conventionally manufactured CRM-containing materials with similar properties. Since the production of HEA parts often requires complicated, expensive, and rare pre-alloyed precursor powders, in situ alloying of pre-mixed elemental powder blends seems the most promising synthesis method for this class of materials. In this regard, molecular dynamics simulation can be used as an efficient predictive tool to investigate the mechanical and deposition properties of HEA and other materials [252]. 

Bulk metallic glasses (BMGs) are a class of materials that can significantly benefit from in situ alloying. As with HEAs, manufacturing of these materials should significantly benefit from the PBF AM production route, as BMGs lose their most attractive properties, such as high corrosion resistance, very special elastic properties, and hardness, if heated above glass transition temperatures and subjected to slow cooling [18,19]. Thus, beam-based AM methods, including PBF ones, providing extreme melting and solidification rates are very promising BMG manufacturing options. Many of the BMG materials introduced for industrial applications contain CRMs [18,253,254]. However, the recent introduction of the BMG compositions without CRM content together with their in situ alloying manufacturing possibilities allow for CRM-sparing manufacturing of these materials [19,254,255,256,257].

Li [258] discussed the prospects of AM routes for the production of HEAs and BMG. SLM is one of the presented possible approaches for HEA fabrication. The author mentioned that SLM of powders blends could be used for HEA synthesizing, including the manufacturing of advanced composite alloys. 

Ocelik et al. [259] synthesized three-layered coatings made of HEA by SLM from pre-mixed elemental powder. The authors found solidification conditions to be the most critical parameter for successful HEA processing, while high-power laser beam with regulated power density and speed is mentioned as a unique advantage of the used additive technology. 

Haase et al. [260] reported on successful simulation and experiments made using SLM of HEA from the blend of elemental powders. The authors mentioned this approach as very attractive due to the ease of modifying target material composition. HEAs produced by this route demonstrated high strength and homogeneous composition. The authors noted the importance of proper adjustment of the laser beam power and scan strategy for obtaining high homogeneity of the final as-printed components. 

Dobbelstein et al. [261] reported on direct metal deposition-assisted synthesis of MoNbTaW refractory HEA. The applied experimental set-up permitted them to perform in situ alloying of the pre-mixed powder blend. The authors also discussed the effect of process parameters on final product oxidation and the formed microstructure and mechanical properties.

Joseph et al. [262] reported on a comparison of microstructure and mechanical properties of the direct laser fabricated (DLF) and arc-melted Al_x_CoCrFeNi HEA. The process and the effect of the production parameters on phase formation, oxidation behavior, and mechanical properties of the final product are described in detail. The authors concluded that the DLF production route permits obtaining materials with the microstructure and properties similar to those obtained by conventional processing, i.e., arc melting.

Cui et al. [22] discussed a thermodynamic approach permitting the prediction of the stability of HEAs. Several examples for successful attempts of laser-based additive manufacturing of these multicomponent alloys from blended elemental Al, Co, Cr, Fe, Ni, and Cu powders are given.

Popov et al. [263] reported on a successful trial synthesizing Al_0.5_CrMoNbTa_0.5_ high-entropy alloy by EBM and comparing the test sample microstructure and homogeneity with those of the material synthesized by a conventional arc melting route. Although the obtained product’s microstructure should be more homogeneous (i.e., the production process must be improved), the authors conclude that there is a principal possibility to synthesize HEAs by AM routes. 

### 4.3. D Laser Shock Peening

As mentioned, the SLM process is a very attractive technology for fabricating components with very complex spatial shapes, such as near-net-shape parts that are impossible or prohibitively complicated to produce through conventional production routes [264]. Indeed, optimization of the AM processing of such components resulted in achieving only slightly lower static mechanical properties than those obtained with conventional processes. However, as with other processes that include layer-by-layer crystallization and solidification, the generation of tensile residual stresses (TRSs) can be considered one of the major deficiencies of many AM methods. In such processes, external energy is supplied with high local density to the last processed layer, leading to the temperature gradients and anisotropic partial annealing of the processed components. Except for the case of EBM, where the build is kept at a strongly elevated temperature, this results in a significant accumulation of TRSs or even considerable component distortion [265]. A considerable effort was devoted to the reduction of TRSs in laser-based technologies, including in situ heating (preheating or high-energy laser re-melting) or post-annealing. This strategy is successful to some degree, and up to 70% reduction of TRS with annealing was reported by Mercelis and Kruth [266]. The main drawback of these methods is that they cannot completely remove either tensile or compressive residual stresses (CRSs). Alternative approaches were tested, and several options emerged: shot peening (SP), laser shock peening (LSP), and 3D laser shock peening (3D LSP) [265]. 

LSP, similar to SP, deforms the surface layer of the part by the application of the shock wave induced by the localized plasma pressure on the component. To enhance this effect, water or solid (glass) confinement is used together with a corresponding laser wavelength setup of 532 and 1064 nm, respectively. It was soon realized that LSP could be performed during the laser-based AM process itself. The 3D LSP process, patented by the Laboratory of Thermomechanical Metallurgy (LMTM) [264], refers to the combination of the SLM process with LSP. LSP treatment is used after several completed layers so that CRS can be handled throughout the component. To achieve this, the LSP setup must be integrated into the SLM device [267]. Several publications proved the benefits of this approach, which can mainly improve the fatigue, wear, and corrosion properties and can also improve geometrical accuracy of parts fabricated by this hybrid manufacturing system, significantly increasing the service life of parts.

Bending fatigue properties of 316L produced by a combination of SLM and 3D LSP (hybrid SLM-LSP) were significantly higher than those of manufactured samples and conventionally produced in both machined and nonmachined conditions [268]. It was shown that by employing 3D LSP, fatigue life is increased by more than 14 times compared to AM samples and by 57 times over that of conventionally produced material. 

It was reported that LSP with solid confinement increases the microhardness near the surface region through the accelerated recrystallization kinetics upon heat treatment, which results in refined equiaxed grains [269].

Kalentics et al. [264] successfully applied the SLM-LSP process for Ti64 alloy bridge-like samples. It was shown that LSP has reduced the distortion angle by up to 75% compared to as-built specimens. Furthermore, 3D LSP was used for nickel-based alloy produced by SLM [270]. A 95% reduction in the number of cracks in this very crack-prone alloy wiring welding has been observed.

Thus, the introduction of laser shock peening is following a general trend of hybrid manufacturing, and the development of additive manufacturing technology is integrating AM with other important technologies, not only into the same processing line but essentially into the same process. 

## 5. Conclusions

Additive manufacturing (AM) technologies are becoming critical to achieving sustainable use of critical raw materials, vital to European industries, in manufacture and repair. The wider introduction of AM technologies in their present shape, the development and incorporation of AM technologies into the hybrid manufacturing production chain, and the development of smart recycling routes for the components using CRMs and CRM-containing alloys are some of the developing trends in sparing utilization of critical materials. The key advantages of AM such as shape optimization and possibilities of on-demand manufacturing present immediate opportunities for the manufacturing industries. The additional opportunity presented by modern AM is in the development of newer compositions with unique properties reducing or even eliminating the use of CRMs. In this aspect, beam-based PBF AM seems to be the most promising technique since it generates unique conditions of fast melting and solidification, beam energy manipulation possibilities for microstructure engineering, in situ alloying, and possibilities of metal–metal and metal–ceramic composite manufacturing. Fast melting and solidification are capable of preserving the unique metastable microstructure of materials, which is not possible with traditional manufacturing methods. This opens wide possibilities for manufacturing materials with unique properties, including high-entropy alloys and bulk metallic glasses, as well as new composite materials for aerospace and biomedical industries. The development of new alloys for AM, specifically targeting preservation of metastable microstructure, already shows possibilities in reducing the consumption of CRMs. An additional benefit of varying beam energy application rates, not only layer by layer but also within each layer, characteristic of beam-based AM promises further possibilities for microstructural and property enhancement along all three dimensions, allowing for material savings.

The environmental friendliness and sustainable nature of AM technologies make further research and development in this area critical for further progress in the fields of “material-oriented manufacturing” and “solid freeform fabrication”.

## Figures and Tables

**Figure 1 materials-14-00909-f001:**
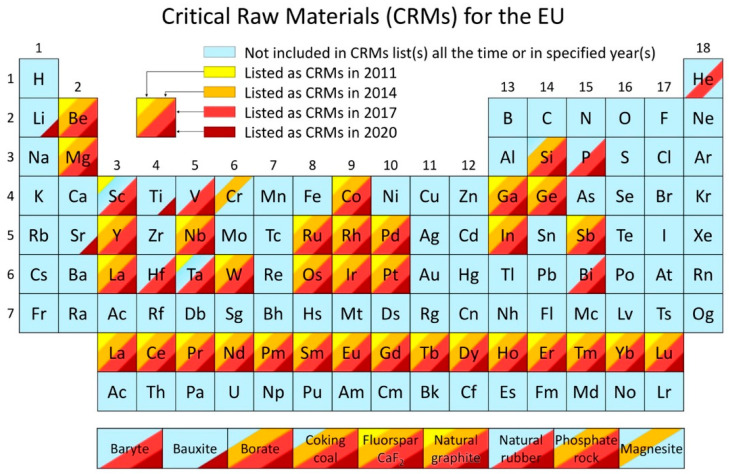
Critical raw materials list for 2011–2020 overlaid on the periodic table of elements (adapted from [8]).

**Figure 2 materials-14-00909-f002:**
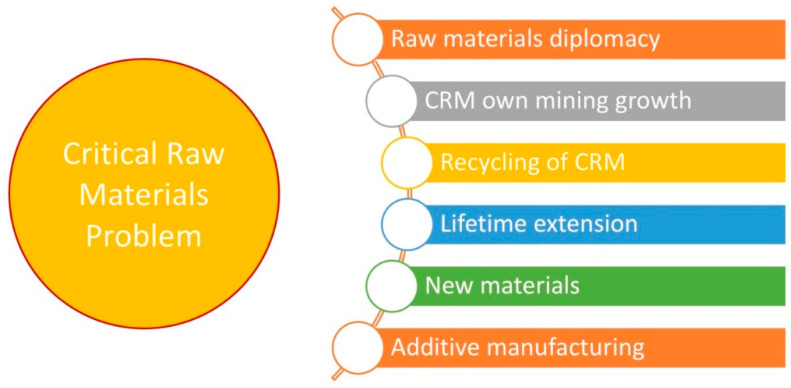
Primary solutions for addressing the issue of critical raw materials.

**Figure 3 materials-14-00909-f003:**
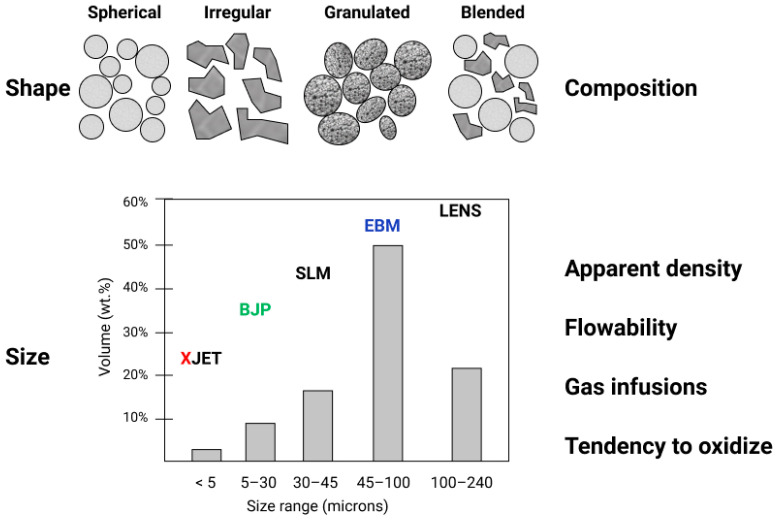
Crucial powder characteristics in additive manufacturing.

**Figure 4 materials-14-00909-f004:**
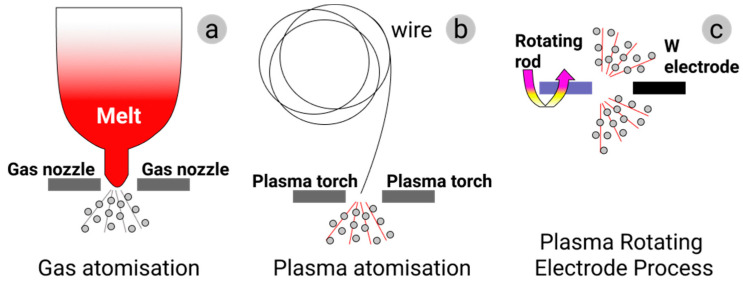
Production of metal powders: (**a**) gas atomization; (**b**) plasma atomization; (**c**) plasma rotating electrode process.

**Figure 5 materials-14-00909-f005:**
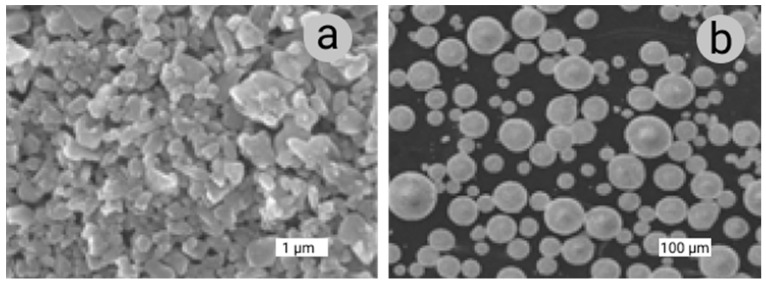
Morphology of powders: (**a**) Al_2_O_3_ submicron powders, mean particle size 0.7 μm, prod. Alcoa A16SG, specific surface area 8.9 m^2^/g, shape factor β = 1.42; (**b**) ZrO_2_ granules TZ-3Y, partially stabilized zirconia powder with a uniform dispersion of 3 mol % yttria, 40 nm, prod. Tosoh Corporation.

**Figure 6 materials-14-00909-f006:**
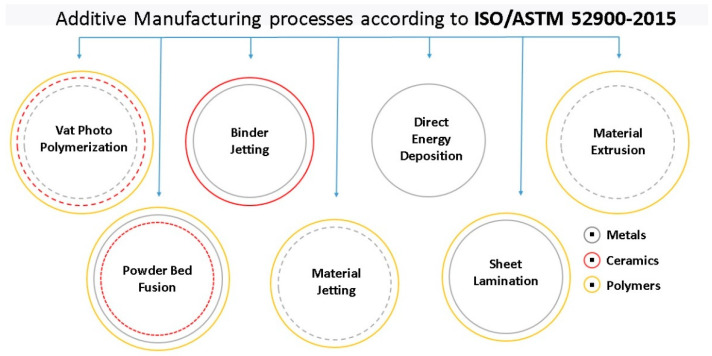
Additive manufacturing processes as defined in [140].

**Table 1 materials-14-00909-t001:** Comparison of most common metal powder bed fusion (PBF) and direct energy deposition (DED) additive manufacturing processes [96,138,151,152,154,156,158,159,160,161,162,163,164,165].

	PBF		DED			
	SLM	EBM	LMD-Powder	LMD-Wire	EBAM-Wire	WAAM
Type of atmosphere	Inert (Ar, N)	Vacuum	Inert	Inert	Vacuum	Inert
Relative part density	≥99.5		≥98	≥98		≥98
Typical layer thickness [μm]	10–100	50–200	10–100 (250–1000)	130–1000	3000	3000
Part complexity	almost unlimited	some limited	limited	Limited	limited	limited
Minimal wall thickness [mm]	≥0.1 mm	≥0.3 mm	≥1 mm	≥1.5 mm	≥1.5 mm	≥1.5 mm
Surface roughness Ra [μm]	5–15	~20	2–91	10–91	20	20–100
Deposition rate [cm^3^/h]	10–25 (100 for multilaser)	3–11 kg/h	<70	100–200 (<500)	100–200 (<500)	100–200 (<500); 1–10 kg/h
Multimaterial	no *	no *	possible	possible	possible	possible
Process energy density [Wh/cm^3^]			17.4	9.2	36	4.9
Energy efficiency [%]	10–20	95		2–5	15–20	<90

* Multi-materials are possible by customized SLM\PBF.

**Table 2 materials-14-00909-t002:** Applied additive manufacturing (AM) technologies for critical raw materials (CRMs) and CRM-containing materials.

CRM	Material Extrusion (Fused Deposition Modelling)	PBF	DED	Vat Photopolymerization	Sheet Lamination	Binder Jetting	Main Application
Ti alloys		[183]	[184]			[185,186]	Aerospace and biomedicine
Ni alloys		[183,187,188]	[188,189]			[190,191]	Aerospace
Al alloys	[192]	[99,193]	[194]		[195,196]		Aerospace
Cu alloys	[192]	[197,198,199]		[200]		[201,202]	Electromagnetic
Mg		[203,204]					Medical
W		[52,53,205]	[206]			[53]	Nuclear reactor
Rare-earth based materials	[207,208]	[209,210]		[210]		[211]	Permanent magnets
Si/SiC/SiO_2_	[212]	[213]		[212]	[214]	[215]	Tooling, optics, medical
Au		[216,217]					Jewelry
Co-Cr alloys		[61,218,219]				[219]	Biomedicine
Nb/Zr/Ta-containing alloys		[220,221]	[90]				Biomedicine
Graphite	[192]		-	[67]	-	[64]	Thermal

## Data Availability

As this is a review paper, no new data was generated for this paper.

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
