# Peer review of "Powder Bed Fusion Additive Manufacturing Using Critical Raw Materials: A Review"

_materials, 2021, doi:10.3390/ma14040909_

Round 1

Reviewer 1 Report

The authors provided the reviews related to CRMs and CRM-containing alloys for current Additive manufacturing (AM) technologies. And the authors discussed how powder-bed AM methods can utilize CRMs in various industries. The strength of this paper is well-written and provides well-defined the CRMs with the relevant AM technologies categorizing the CRMs and AM methods by detailing specifics of each method and requirements for fabrications. The manuscript is recommended for publication after addressing the following concerns:

1.The authors need to address the proposed review approach or method in details. For example, which scientific databases/repositories have been considered for the proposed review method. And, it could be interesting to provide the information of the geographical areas distribution in terms of CRMs and AM technologies.

2.The authors need to provide and add industrial case studies, applications or components (parts) from the reviews in Table 2.

3.The regarding AM-related possibilities to reduce and optimize the use of CRMs and CRM- based materials, the authors need to add the discussion of Design for AM and Topology optimization approaches.

Author Response

We want to thank the Reviewer for his time and professional help.

The following revisions have been performed:

1. We have added several paragraphs about review methodology in the Introduction (lines 137-161).
2. We have added a new column “Main application”  in Table 2.
3. Few sentences are now added which may be checked by seeing lines 776-782

Reviewer 2 Report

In this manuscript, the authors have placed their attention on a review on powder bed fusion additive manufacturing using CRM materials.

The work is well structured and the proposed goals were achieved. I think that this work could be of interest for the field of materials engineering and material science. The paper is technically sound. Many topics have been treated with care. The Authors carried out the bibliographic research with care and attention. The manuscript is very interesting overall. The manuscript contains new information to justify publication.

-Pay attention to the language: the whole manuscript requires a thorough English revision, some sections are hard to read, and it becomes quite hard to understand the actual message from the authors. 

Furthermore, I have not encountered any major problems. 

Specific Comments

- Some sentences are too long. I ask the authors to break them.

- Keywords: replace “powders for AM” with “powders for Additive Manufacturing”

- Fig 1: The Authors should better describe Figure 1 within the text. I suggest making clear use of colors in reference to the year and chemical elements. The vision of the figure alone may be unclear.

- Line 89-90: The authors write “A simple search through research databases for the keywords, critical raw materials” gives 453 publications in the Scopus database alone”. Here too, indicate the time interval more clearly.

- Lines 200, 216, 509 and other: write "AM" in full,

- Sections 2.2.1, 2.2.2 and 2.2.3: This reviewer recommends adding any pictures about the technique. This makes the manuscript lighter to read and immediately makes the reader understand the type of technique.

- Line 315: Add the chemical symbol for tungsten.

- Line 322: Superscript "2".

- Lines 516, 536 and other: Pay attention to subscripts.

- Fig.4: The provenance of this figure is not clear.

- Section 3: I suggest the authors add a small paragraph on implementation approaches: bottom-up or top down techniques.

- Table: I suggest the authors to clearly express the bibliography within the table.

 - I suggest that the authors reread the whole manuscript: there are small typos.

- Update the dates of last access to the websites.

- It is not clear what reference number 41 is; same comment for 141.

Best Regards

Author Response

We want to thank the Reviewer for his valuable comments.

The necessary revisions have been performed, please see the attached point-by-point response to the comments.
